# A Systematic Review on Intervention Treatment in Pathological Gambling

**DOI:** 10.3390/ijerph21030346

**Published:** 2024-03-14

**Authors:** Diana Moreira, Paulo Dias, Andreia Azeredo, Anabela Rodrigues, Ângela Leite

**Affiliations:** 1Centro de Solidariedade de Braga/Projecto Homem, 4700-024 Braga, Portugal; dimoreira@ucp.pt (D.M.); pcdias@ucp.pt (P.D.); andreia.azeredo@projectohomem-braga.pt (A.A.); 2Centre for Philosophical and Humanistic Studies, Faculty of Philosophy and Social Sciences, Universidade Católica Portuguesa, 4710-362 Braga, Portugal; anabela.rodrigues@ucp.pt; 3CPUP—Center for Psychology at University of Porto, Faculty of Psychology and Educational Sciences, University of Porto, 4099-002 Porto, Portugal; 4Institute of Psychology and Neuropsychology of Porto—IPNP Health, 4000-055 Porto, Portugal; 5Observatory Permanent Violence and Crime (OPVC), FP-I3ID, 4249-004 Porto, Portugal

**Keywords:** behavioral addiction, pathological gambling, intervention treatment, systematic review

## Abstract

Over the last century, there has been a growing interest in researching pathological gambling, particularly in industrialized nations. Historically, gambling was widely perceived as morally questionable, condemned by religious groups. However, contemporary concerns have shifted towards the health repercussions of gambling disorders and broader societal impacts like increased crime and money laundering. Governments, aiming to mitigate social harm, often regulate or directly oversee gambling activities. The global surge in legal gambling has resulted in a substantial rise in its prevalence, popularity, and accessibility in the last two decades. This paper provides a comprehensive overview of global research on interventions for pathological gambling. Through a systematic search on platforms such as EBSCO, PubMed, and Web of Science, 13 relevant records were identified. The revised findings indicate a heightened occurrence of behavioral addictions, linking them to the early onset of gambling issues and their severe consequences. The research emphasizes the active role that clients play in the process of self-directed change and therapy. Therapists recognizing clients as both catalysts for change and potential obstacles can enhance their effectiveness. A common source of resistance arises when clients and therapists are in different stages of the change process, underlining the importance of therapists aligning with clients’ readiness for change. Recognizing the urgent need for a better understanding of this problem in adolescents, this study emphasizes the necessity to tailor prevention and treatment plans based on gender and age-specific requirements.

## 1. Introduction

The objective of this study is to conduct a comprehensive survey of global research on pathological gambling. The study aims to address the following research inquiries: What interventions and methodologies of study have been prominent in gambling research? What implications and recommendations for policy and further research can be derived from these findings to inform people who are in practice?

Gambling is defined as the act of taking a certain risk on the outcome of an event determined by chance [1], and for many people around the world, gambling is often viewed as a fun and relatively not dangerous activity. It is considered an acceptable and popular social activity and leisure activity in many cultures. Gambling is a common pastime in most cultures. These include, but are not limited to, lottery games, sports betting, slot machine gambling, and casino gambling, where one can participate in physical (or offline) or online gambling. While gambling is an exciting and social leisure activity for most people, a minority of players have difficulty controlling their gambling behavior, resulting in gambling-related problems. Gambling problems may involve multiple domains, including but not limited to financial problems, psychological and emotional distress, and adverse functional consequences, such as relationship problems or unemployment [2,3].

Epidemiological studies estimate that the prevalence of pathological gambling among adults in the past year ranges from 1.1% to 3.5% [4,5,6], although differences between studies may be partly attributable to sampling and measurement artifacts [7]. Although the terms “problem gambling” and “pathological gambling” are used differently, problem gambling is often used to describe the intermediate or subclinical form of the “pathological gambling” disorder. Problem gambling and pathological gambling are serious public and mental health problems that affect individuals, families, and communities [5]. Problematic and pathological gambling is associated with impaired mental functioning, reduced quality of life, legal problems, and high rates of bankruptcy, divorce, and incarceration [8,9]. Problem gambling is also associated with other mental disorders, including depression, anxiety disorders, bipolar disorder, personality disorders, and alcohol, drug, and nicotine use [10,11].

Therefore, pathological gambling is commonly seen as a social behavior. Quilty et al. [12] characterized it as an activity that harms or disrupts various aspects of life, affecting individuals, families, and friendships, with a global prevalence of 2.3% [13]. The prevalence of gambling varies across different regions of the world. In the United States, 70–90% of adults engage in gambling at some point in their lives [14,15]. Research conducted in various countries indicates that the prevalence of problematic gambling in the 12 months prior to the study ranged from 0.3% in Sweden to 5.3% in Hong Kong [16].

For many years, scientists have explored the connection between gambling and involvement in criminal activities. Until 2012, the DSM-IV-TR utilized a criterion involving engagement in illegal actions for gambling purposes to diagnose pathological gambling. This criterion specifically mentioned committing illegal acts like forgery, fraud, theft, or embezzlement to fund gambling [17]. However, in 2013, the American Psychiatric Association removed this criterion in the DSM-5 [18]. Despite this change, various studies [19,20,21,22] have demonstrated the correlation between gambling and criminal activity. Consequently, individuals who are both incarcerated gambling addicts and repeat offenders involved in gambling-related crimes continue to be imprisoned [23,24,25,26].

In the past, there has been a higher prevalence of gambling behavior among men compared to women, as indicated by research [4]. This suggests that men are more inclined to gamble and encounter issues related to gambling compared to women, as highlighted by the National Research Council in 1999. Societal norms and attitudes regarding the acceptance or non-acceptance of male and female gamblers may have contributed to these gender disparities. However, efforts promoting gender equality have played a role in narrowing this gap over time.

Several psychological interventions have been described for treating pathological gambling, including psychodynamic interventions, Gamblers Anonymous, inpatient rehabilitation programs, behavioral interventions, cognitive interventions, and cognitive behavioral interventions [27]. Cognitive and behavioral interventions have been recommended as “best practice” in treating pathological gambling. Behavioral interventions using a range of techniques are the most evaluated approaches in the psychotherapy of pathological gambling. The recent literature evaluating behavioral treatments has shifted away from aversion therapies and toward alternative behavioral techniques, such as interventions based on desensitization and exposure procedures. A variety of other behavioral techniques, such as alternative activity scheduling, problem-solving training, financial planning and boundary setting, social skills training, and relapse prevention, have been incorporated into standardized cognitive behavioral programs as treatment components [28,29].

In fact, behavioral interventions, using a range of techniques, have been the most common approach to the psychological treatment of pathological gambling. In accordance with learning principles, behavioral approaches to the treatment of pathological gambling have commonly applied classical and operant conditioning techniques to reduce the arousal and excitement associated with gambling [30,31]. Many studies have evaluated aversive techniques either in isolation [32,33,34] or in combination with other behavioral procedures, such as supportive therapy, covert sensitization, positive reinforcement, exposure techniques, and stimulus control techniques [35,36]. Although these studies indicate that aversive therapy, both alone and in combination with other techniques, generally produce moderate improvements in gambling behavior, it is argued that it is difficult to ethically justify the use of a procedure that has been criticized as an intrusive, unpleasant, and dehumanizing procedure that causes undue emotional distress [30,37].

For this reason, the literature evaluating behavioral treatment shifted from aversive therapy to alternative behavioral techniques, such as behavioral counselling [38,39], imaginal relaxation [33,40] and desensitization and exposure techniques [41,42,43,44,45,46]. Behavioral techniques that have been employed as treatment components of standardized treatment programs for pathological gambling include alternative activity planning, problem solving training, financial planning and limit setting, social skills and communication training, relapse prevention, stimulus control, in vivo exposure, and imaginal desensitization [28,29,47,48,49].

One of the most striking features of gambling disorder is gaming-related cognitive distortions [50], such as illusion of control (i.e., the ability to control and predict wins), selective win retrieval, loss minimization, and rational thinking [51,52]. These dysfunctional gambling-related cognitions and impulsivity play important roles in the development, maintenance, and severity of gambling disorders [53,54,55,56]. Earlier studies have shown a reciprocal connection between the intensity of gaming disorder and cognitive biases related to gaming [57,58]. However, Mallorquí-Bagué et al. [55] discovered no link between cognitive distortions and the severity of gambling disorder.

Various studies, adopting different perspectives, have investigated the efficacy of intervention strategies in addressing problematic gambling. One example is the proposal of “forced breaks” as a means to regulate impulsive gambling, under the assumption that some gamblers may experience a sense of helplessness or detachment when confronting addiction (e.g., Broda et al. [59]). Additionally, alternative approaches have been explored, such as implementing a restricted budget for discretionary spending [60], utilizing behavioral tracking tools [61], employing push messages [62], incorporating human–computer interaction and behavioral-science-enhanced intervention tools [63], and promoting responsible marketing [64].

However, the majority of tools align with Blaszczynski et al.’s [64] concept of a “gambling operator-based approach”, designed with a more specific purpose and audience in mind. However, as casinos expand their operations to the online domain, the involvement of third-party entities in gambling activities, such as payment gateways (digital platforms facilitating online financial transactions), has increased. The existing literature indicates that “payment institutions serving gambling customers lack comprehensive policies and strategies” to assist users who are financially vulnerable [65,66]. Despite the proliferation of strategic intervention tools (e.g., Collins et al. [67]; Hansen and Rossow, [68]), there is a scarcity of research focused on the utilization of “non-operator” tools for harm reduction.

Moreover, research within this thematic area often gathers data from gambling operators or utilizes Responsible Gambling tools that specifically target operator sites. These studies typically concentrate on identifying “risk groups” within the gambling population [69,70]. Consequently, there is a pressing need to delve into research that focuses on the economic conditions of the general population of gambling users [71,72].

## 2. Methods

### 2.1. Search Strategy

Studies were identified through search on EBSCO, PubMed, and Web of Science, with the limit of 29 September 2023. The reference lists of the selected studies were also reviewed to identify other relevant studies. The search equation in EBSCO was as follows: AB (“gambling disorder” OR “gambling addiction” OR “pathological gambling” OR “problem gambling” OR gambl*) AND AB (“therap* communit*” OR “therap* communit* for substance abuse” OR institution* OR organization*) AND AB (intervention* OR program* OR prevention OR strateg* OR “best practices” OR treatment OR therapy OR care), in Pubmed ((“gambling disorder” [Title/Abstract] OR “gambling addiction” [Title/Abstract] OR “pathological gambling” [Title/Abstract] OR “problem gambling” [Title/Abstract] OR gambl* [Title/Abstract]) AND (“therap* communit*” [Title/Abstract] OR “therap* communit* for substance abuse” [Title/Abstract] OR institution* [Title/Abstract] OR organization* [Title/Abstract])) AND (intervention* [Title/Abstract] OR program* [Title/Abstract] OR prevention [Title/Abstract] OR strateg* [Title/Abstract] OR “best practices” [Title/Abstract] OR treatment [Title/Abstract] OR therapy [Title/Abstract] OR care [Title/Abstract]), and in Web of Science ((AB = (“gambling disorder” OR “gambling addiction” OR “pathological gambling” OR “problem gambling” OR gambl*)) AND AB = (“therap* communit*” OR “therap* communit* for substance abuse” OR institution* OR organization*)) AND AB = (intervention* OR program* OR prevention OR strateg* OR “best practices” OR treatment OR therapy OR care).

### 2.2. Study Selection

The selection of studies for analysis adhered to specific inclusion and exclusion criteria. Inclusion criteria encompassed studies that were (a) empirical, (b) involved participants aged 12 years and above*, and (c) focused on interventions for gambling disorder. Exclusion criteria involved: (1) wrong publication type—case studies, book chapters, theoretical essays, and systematic reviews with or without meta-analysis; (2) wrong theme—studies unrelated to intervention in gambling disorder; (3) wrong population—studies not involving a participant group aged 12 years or older; (4) wrong outcome—studies lacking an intervention component.

According to Berger [73], adolescence begins at about 12/13 years of age, or even earlier. It is a phase in which individuals seek to assert themselves and establish their identity alongside their family and/or peer group. Therefore, behaviors of opposition and rebelliousness, marked by a greater or lesser degree of aggression, are normative and frequent [74]. However, and not infrequently, these may assume contours of some severity [75]. This phase is marked by some behaviors typical of adolescence (e.g., use of psychoactive substances, appetite for risk, oppositional behaviors) [76]. Additionally, social norms tend to impose a set of demands (e.g., school, and academic success) and options (e.g., entry into higher education, compatible professional activity), to which young people are not always prepared to respond appropriately [77].

The search was restricted based on language (Portuguese, English, Spanish, or French), and duplicate articles were removed from consideration.

Two reviewers (DM and AA), working independently, selected the studies based on their titles and abstracts, adhering to the guidelines outlined in the PRISMA framework [78,79].

EndNote X9^®^ and Rayyan^®^ were used, considering the manual check of duplicate articles as a reference. A study was considered a duplicate when a bibliographic record (authorship, title, periodical, number, volume, number of pages) was retrieved more than once in one or more electronic databases, regardless of whether it presents abbreviations and variations in the spelling of any term [80]. When a reference presented the same title but had some missing data or misspelled terms, such as volume and page number [81], it was considered a duplicate abstract if it presented the same content.

EndNote X9^®^ (version number 9) is a reference manager software that compares author, year, title, and publication type to identify duplicate articles [82]. The online version was used because it is freely available and is frequently used in the production of systematic review studies, although this use is not always reported in published articles [83].

Rayyan^®^ is a specific computational tool for producing systematic reviews [84], with scientific reports of good results in automatically identifying true duplicate articles [85]. Unlike other tools used to aid systematic reviews, the elimination of duplicates is carried out after checking by the researcher.

The level of agreement in the study selection process was evaluated using Cohen’s Kappa, indicating an almost perfect agreement at *K* = 0.98, *p* < 0.001 [86]. Any discrepancies among the reviewers were addressed through discussion and resolved by reaching a consensus.

### 2.3. Identification and Screening

A total of 627 studies, spanning the period from 1931 to 2023, were initially identified through database searches. Among them, 253 duplicate studies were removed. The titles and abstracts of the remaining 374 studies underwent evaluation, and 78 of these were selected for comprehensive full-text analysis. Meanwhile, 296 articles were excluded based on the following criteria: incorrect publication type (*n* = 176), unrelated theme (*n* = 104), and inappropriate outcome (*n* = 16). Following the full-text analysis, twelve articles were ultimately retained for inclusion in this review, along with one manually identified article (Figure 1). In total, 13 articles were considered, and pertinent information such as objectives, sample details (age, % male, and type of sample), intervention program details (program and total duration), results, and main conclusions were extracted from each study.

## 3. Results

In the current cultural and therapeutic landscape emphasizing the rise in integrative therapeutic approaches, Transtheoretical Therapy emerges as an alternative. This approach has gained traction as it aligns with the increasing popularity of integrative therapeutic systems. Through a comparative analysis of 18 prominent therapeutic systems, five fundamental change processes have been identified (Table 1). These processes are adaptable to both individual settings and experiential levels. Examining how individuals undergo change through formal therapy has revealed four distinct stages of change. Individuals who underwent therapeutic changes, as opposed to those who did not, demonstrated the use of three verbal change processes during the contemplation and decision phases. Additionally, two behavioral processes were observed during the action and maintenance phases. While the verbal process is not theoretically incompatible, the crucial aspect lies in preparing the client for action. Once the client commits to action, the behavioral process takes precedence, as emphasized by Prochaska and Di Clemente [87] (Table 1).

Hodgins et al., conducted a series of studies focusing on the treatment of problem gambling. In a randomized study, two brief interventions for problem gambling were compared to a control group on a waiting list. During the 12-month follow-up period, 84% of participants (*N* = 102) reported a significant reduction in their gambling behavior. Notably, participants who received an intervention involving motivational phone calls and a self-help workbook delivered by mail demonstrated better outcomes compared to those who only received the workbook, particularly when compared to a one-month waitlist control. Results indicated that, at the 3- and 6-month follow-ups, participants who underwent the motivational talk and received the workbook showed superior progress compared to those who solely received the workbook. However, by the 12-month follow-up, the benefits of motivational interviewing and workbook intervention were evident primarily in participants with less severe gambling problems. In summary, these findings affirm the effectiveness of straightforward telephone and postal interventions for addressing gambling addiction [88] (Table 1).

Furthermore, a 24-month follow-up from a randomized clinical trial in 2004, involving two brief treatments for problem gambling (*N* = 67), revealed significant advantages for participants who received a motivational intervention in addition to a workbook compared to those who exclusively received the workbook. Although there was no notable difference in the number of participants reporting 6 months of alcohol abstinence between the two groups, the motivational intervention group exhibited fewer gambling days, lower financial losses, and lower scores on the South Oak Gambling Screen. They were also more likely to be rated as showing improvement compared to the group relying solely on self-help workbooks. In summary, these findings provide further support for the efficacy of brief telephone and postal interventions for individuals dealing with problem gambling [89] (Table 1).

Five years later, a randomized clinical trial was conducted to assess the effectiveness of brief treatment in individuals with pathological gambling recruited through media channels (*N* = 314). The trial compared two self-directed motivational interventions with a 6-week waitlist control and a control group receiving only workbooks. The brief motivational therapy involved motivational phone interviews and self-help workbooks sent by mail, while brief motivational review therapy included motivational phone interviews, workbooks, and six review calls over a nine-month period. The main outcome measures were gambling frequency and dollar losses. As anticipated, participants receiving short-term interventions and booster doses reported reduced gambling compared to those in the control group after 6 weeks. Over the first 6 months of follow-up, participants in the short-term and short-term booster groups gambled significantly less frequently than those in the workbook-only group. However, participants in the workbook-only group significantly reduced their losses over the year and no longer met the criteria for pathological gambling. Contrary to expectations, participants in the short-term review group did not exhibit greater improvement than those in the short-term treatment group. In conclusion, these findings endorse the effectiveness of brief motivational therapy in addressing pathological gambling [90] (Table 1).

Motivational interviewing (MI) emerges as a promising brief intervention for individuals seeking to diminish or cease their gambling activities. In a randomized clinical trial, researchers examined the effectiveness of a single face-to-face MI session compared to a controlled interview (CI) in mitigating gambling behavior among individuals expressing concerns about their gambling habits. Following an intervention, a 12-month follow-up involving 81 participants recruited through media channels was conducted at intervals of 1, 3, 6, and 12 months. At the conclusion of the 12-month intervention period, participants who underwent MI exhibited significantly lower monthly expenditures on gambling and fewer gambling days per month; they also reported significantly lower levels of stress compared to those who underwent CI. Notably, both groups displayed an overall reduction in the severity of gambling-related problems [91] (Table 1).

A group of individuals with gambling problems (*N* = 180) were randomly assigned to one of four conditions: assessment alone, 10 min brief counseling, one session of motivation-enhancing therapy (MET), or one session of MET coupled with three sessions of cognitive behavioral therapy. Gambling behavior was assessed at baseline, 6 weeks, and 9 months of follow-up. In comparison to assessment alone, only brief counseling demonstrated a significant reduction in gambling between baseline and week 6, and it was linked to a clinically noteworthy decrease in gambling at the 9-month mark. Between weeks 6 and 9, MET plus cognitive behavioral therapy exhibited a substantial reduction in gambling compared to the control condition. These findings suggest that extremely brief interventions can be effective in decreasing gambling tendencies among individuals with problem and pathological gambling behavior who are not actively seeking formal treatment [92] (Table 1).

One of the prominent addiction support organizations in the Netherlands has initiated a pilot program to explore the potential of utilizing existing treatment approaches grounded in cognitive behavioral therapy and Motivational Interviewing (MI), referred to as “lifestyle training”, for addressing internet addiction. A study was conducted to assess this pilot treatment program by qualitatively examining the experiences of therapists working with 12 individuals self-identifying as Internet addicts. Although commonly employed for drug addiction and pathological gambling, therapists found this program to be well-suited for addressing issues related to Internet addiction. The interventions primarily focused on regulating and reducing Internet usage, emphasizing the improvement of (physical) social connections, re-establishing a proper daily routine, constructive utilization of free time, and addressing distorted beliefs. Therapists reported that all 12 patients undergoing treatment demonstrated progress, with patients indicating improvements in treatment satisfaction and actual behavioral changes [93] (Table 1).

Furthermore, a study investigated the efficacy of three brief telephone interventions in comparison to standard helplines for assisting individuals in reducing gambling habits: (1) Personal Motivational Interview (MI); (2) Personal Motivational Interview plus Recognition Knowledge-Behavior Self-Help Workbook (MI + W); and (3) Single Motivational Interview Plus Exercise Book Plus Four Telephone Follow-Up Interviews (MI + W + B). The control group received the standard hotline service (TAU). The follow-up assessments, conducted in a blinded manner after 3, 6, and 12 months, involved 462 adults with gambling problems in a randomized clinical trial. Interestingly, the results indicated no significant differences between the treatment groups, despite participants demonstrating substantial reductions in gambling throughout the 12-month follow-up period. However, subgroup analyses revealed that MI + W + B led to improvements in the number of days spent gambling and the amount of money lost, surpassing the outcomes achieved with MI or MI + W. Additionally, MI + W + B showed enhanced risk factor improvements for race, gambling severity, and psychological distress related to lost dollars (all *p* < 0.01). TAU and MI were deemed equivalent in terms of dollars lost [94] (Table 1).

A study involving 146 individuals providing gambling support counseling to Australian not-for-profit organizations was conducted [95]. Fifty-five percent of the participants completed the 18-month follow-up. The results from a multilevel model revealed a significant reduction in the severity of participants’ gambling problems. This reduction was characterized by a small change in effect size in the short term and a substantial change in effect size observed at the 18-month follow-up. Interestingly, the study found that the behavior of practitioners adhering to Motivational Interviewing (MI) principles did not significantly predict improvements in participants’ gambling problem severity. On the other hand, the behavior of practitioners not adhering to MI principles was identified as a significant predictor of worsened participant gambling problem severity. This study emphasizes the importance of eliminating practitioner behaviors inconsistent with MI, such as confrontation and persuasion, to prevent unfavorable treatment outcomes [95] (Table 1).

Numerous studies have emphasized the importance of considering the clinical implications of neurocognitive impairment in individuals seeking treatment for cocaine use. One particular study proposed that impaired decision making might contribute to an increased risk of treatment discontinuation among individuals dependent on cocaine (CDI). To investigate this hypothesis, the study compared the baseline performance of CDI individuals, who either completed or discontinued treatment in an inpatient community, using two validated decision-making tasks—the Iowa Gambling Task (IGT) and the Cambridge Gambling Task (CGT). The results indicated that, in contrast to patients who successfully completed treatment, CDI individuals with early treatment cessation did not consistently demonstrate a favorable response pattern in IGT progression and exhibited a poorer ability to select the most likely outcome in CGT. Interestingly, there were no significant group differences in betting behavior. The findings suggest that incorporating neurocognitive rehabilitation targeting poor decision making could yield clinical benefits for individuals with CDI enrolled in long-term inpatient care programs [96] (Table 1).

In order to assess the effectiveness of an internet-based cognitive behavioral therapy (I-CBT) program for treating problem gambling, a study compared it with a waiting list control and an active comparison condition involving monitoring, feedback, and support (I-MFS). A total of 174 participants were randomly assigned to three conditions, and the study focused on variables such as gambling outcomes and related mental health measures. Participants in both active conditions (I-CBT and I-MFS) engaged in six online modules. Both I-CBT and I-MFS demonstrated significant treatment benefits in reducing gambling severity. However, I-CBT not only led to improvements in various gambling-related aspects but also showed positive effects on psychological outcomes. In comparison to I-MFS, I-CBT had a more substantial impact on seven outcome measures related to play instinct, cognition, stress, and life satisfaction. Participants in the I-CBT group also reported higher satisfaction with the program. The treatment gains observed in both active conditions remained stable at the 12-month follow-up. These findings suggest that the benefits of I-CBT go beyond the nonspecific effects of engaging in online treatment or receiving motivation, feedback, and support. Online treatments for gambling-related issues may prove valuable in enhancing help-seeking behaviors and treatment engagement, making them suitable for integration into stepped care approaches for treatment [97] (Table 1).

Personalized normative feedback (PNF) is a brief intervention designed to correct misconceptions about the prevalence of certain behaviors by presenting individuals engaging in such behaviors with information that their own behavior is atypical compared to actual norms. In a recent randomized controlled trial focused on college students with gambling issues, a computer-assisted PNF intervention was evaluated. Following the baseline assessment, 252 second-year students with an SOGS score of 2 or more were randomly assigned to receive either PNF or attention control feedback. Follow-up assessments were conducted at 3- and 6-months post-intervention. The results demonstrated a significant intervention effect in reducing the perceived standard of winning and losing amounts, as well as decreasing the actual amount lost and gambling problems at the 3-month follow-up. All intervention effects persisted at the 6-month follow-up, except for the reduction in gambling problems. Mediation results indicated that changes in perceived norms at 3 months mediated the intervention effects. Furthermore, the effectiveness of the intervention was influenced by self-identification with other student gamblers, suggesting that PNF was more effective in reducing gambling behaviors for those who strongly identified with their fellow student gamblers [98] (Table 1).

In 2016, Greenland implemented a novel addiction treatment service targeting alcohol, cannabis, and gambling addictions within the community. This service involved establishing treatment centers in each of the five communities and collaborating with a central private treatment provider to offer services to individuals in areas without local treatment centers. Substantial individual variations were observed between those opting for topical and intensive therapy. Women with young children and employed women showed a higher likelihood of receiving local treatment, with their alcohol consumption primarily concentrated on weekends and holidays. On the other hand, individuals opting for intensive treatment were more evenly distributed between men and women, had fewer minor children at home, exhibited more pronounced patterns of heavy drinking, and were more frequent marijuana users. These findings align with the expectation that local treatment is more appealing to individuals with domestic responsibilities. The study emphasizes the importance of considering population differences when planning treatment services, recognizing that diverse populations may have distinct needs [99] (Table 1).

**Table 1 ijerph-21-00346-t001:** Summary of Studies’ Characteristics.

Study Identification	Objectives	Sample			Intervention Programs		Results and Main Conclusions
		Age	% Male	Type of Sample	Program	Total Duration	
Abbott et al. (2017) [94]	Investigated three brief telephone interventions to determine whether they were more effective than standard helpline treatment in helping people reduce gambling.	**TAU** *M* = 40.30 *SD* = 13.60 **MI** *M* = 39.10 *SD* = 13.10 **MI + W** *M* = 39.90 *SD* = 11.70 **MI + W + B** *M* = 37.50 *SD* = 13.10	**TAU** 41 **MI** 47 **MI + W** 45 **MI + W + B** 55	*N =* 462 TAU—*n* = 112 MI—*n* = 112 MI + W—*n* = 118 MI + W + B—*n* = 116	AUDIT-C EGM K-10 PGSI	3, 6, and 12 months.	• There were no differences across treatment arms although participants showed large reductions in gambling over the 12-month follow-up period. • Motivational interview (MI) + cognitive behavioral self-help workbook + follow-up telephone interviews were associated with greater treatment goal success for higher gambling severity than helpline standard care (TAU) or MI at 12 months and better for those with higher psychological distress and lower self-efficacy to MI. • TAU and MI were found to be equivalent in terms of dollars lost.
Casey et al. (2017) [97]	To determine whether the pre-treatment variables were predictive of treatment outcomes in the two treatment conditions.	**I-CBT** *M* = 44.82 *SD* = 9.02 **I-MFS** *M* = 44.08 *SD* = 10.48 **Waitlist Control** *M* = 44.18 *SD* = 9.51	**I-CBT** 42 **I-MFS** 39 **Waitlist Control** 42	*N* = 174	GSAS GUS GRSEQ GRCS DASS-21 AUDIT Brief COPE QOLI SWLQ	12 months	• Both internet-based cognitive behavioral therapy program and monitoring, feedback, and support conditions resulted in significant treatment gains on gambling severity. • However, internet-based cognitive behavioral therapy program was also associated with reductions in a range of other gambling-related and mental health outcomes.
Diskin & Hodgins (2009) [91]	To explore the effect of a motivational interviewing approach over and above generic therapist contact.	*M* = 45.00 *SD* = 10.60	57	*N* = 81	PH1-PRIME-MD PRIME-MD DAST NODS SOGS BSI 5 statements describing the interaction with the therapist	1, 3, 6 and 12 months	• At 12 months post-intervention participants in the motivational interviewing condition spent significantly less money on gambling per month, gambled fewer days per month, and reported significantly less distress than participants in the control interview condition.
Flyger et al. (2020) [99]	Reveal data on treatment for substance abuse of adults receiving treatment in their local area or in a central treatment facility in Greenland in 2016–2017.	**Local** *M* = 37.78 *SD* = 11.82 **Central** *M* = 36.91 *SD* = 11.92	**Local** 66 **Central** 56	*N* = 445 Local—*n* = 189 Central—*n* = 256	AUDIT DUDIT ASI	ASI is provided at 6 weeks, the end of treatment, and 3-, 6- and 12-months post-treatment.	• Individuals in local treatment are more often women with minor children and a job, and their alcohol use is concentrated on weekends/holidays. • Individuals in central treatment are more equal in both genders, few have minor children living at home, heavy drinking is more pronounced, and cannabis is used more frequently as well.
Hodgins et al. (2001) [88]	Two brief treatments for problem gambling were compared with a waiting-list control in a randomized trial.	*M* = 46.00 *SD* = 9.00	48	*N* = 102	SOGS DSM-III	12 months	• Participants who received a motivational enhancement telephone intervention and a self-help workbook in the mail, but not those who received the workbook only, had better outcomes than participants in a 1-month waiting-list control. • Participants who received the motivational interview and workbook showed better outcomes than those receiving the workbook only at 3- and 6-month follow-ups. • At the 12-month follow-up, the advantage of the motivational interview and workbook condition was found only for participants with less severe gambling problems.
Hodgins et al. (2004) [89]	A 24-month follow-up of a randomized clinical trial of 2 brief treatments for problem gambling revealed an advantage for participants who received a motivational telephone intervention plus a self-help workbook compared with participants who received only the workbook.	*M* = 46.00 *SD* = 10.00	44	*N* = 67	SOGS	7–12 13–18 19–24 months	• A 24-month follow-up of a randomized clinical trial of 2 brief treatments for problem gambling revealed an advantage for participants who received a motivational telephone intervention plus a self-help workbook compared with participants who received only the workbook.
Hodgins et al. (2009) [90]	The efficacy of brief treatments for media-recruited pathological gamblers was tested in a randomized clinical trial design.	**Brief Treatment** *M* = 40.30 *SD* = 11.30 **Brief Booster Treatment** *M* = 41.40 *SD* = 11.40 **Workbook Only** *M* = 39.90 *SD* = 12.00 **Waitlist Control** *M* = 39.80 *SD* = 12.00	**Brief Treatment** 45 **Brief Booster Treatment** 44 **Workbook Only** 45 **Waitlist Control** 45	*N =* 314 **Brief Treatment**—*n* = 83 **Brief Booster Treatment**—*n* = 84 **Workbook Only**—*n* = 82 **Waitlist Control**—*n* = 65	DSM–IV PGSI-CPGI NODS SOGS GASS	32-months	• Brief and brief booster treatment participants reported less gambling at 6 weeks than those assigned to the control groups. • Brief and brief booster treatment participants gambled significantly less often over the first 6 months of the follow-up than workbook only participants.
Milic et al. (2021) [95]	To investigate the effectiveness of MI on the outcomes for help-seeking problem gamblers when delivered by practitioners in routine practice at a community-based GHS.	*M* = 42.00 *SD* = 13.50	75	*N* = 146	9-item Problem Gambling Severity Index K-10 ASI-G	Time 1 assessment over the phone before the MI session, time 2 assessment at 1–2 weeks after the MI session, time 3 at 6–8 weeks, time 4 at 6 months and time 5 at 18 months after the MI session.	• A significant reduction in participants’ problem gambling severity and psychological distress was evidenced, which was a small effect size change in the short-term and large effect size change by the 18 months follow-up. • MI non-adherent practitioner behaviors were significant predictors of deterioration in participants’ problem gambling severity and psychological distress.
Neighbors et al. (2015) [98]	To evaluate the efficacy of a PNF intervention for problem gambling college students.	*M* = 23.11 *SD* = 5.34	60	*N* = 252	SOGS Gambling Quantity and Perceived Norms Scale Gambling Problems Index Measure of Identification with Groups	3 and 6 months	• Significant intervention effects in reducing perceived norms for quantities lost and won, and in reducing actual quantity lost and gambling problems at the 3-month follow-up. • All intervention effects except reduced gambling problems remained at the 6-month follow-up.
Petry et al. (2008) [92]	To evaluate the efficacy of three brief interventions.	**Control** *M* = 41.40 *SD* = 12.50 **Brief Advice** *M* = 43.5 *SD* = 14.40 **MET** *M* = 45.00 *SD* = 13.80 **MET + CBT** *M* = 44.00 *SD* = 10.20	**Control** 69 **Brief Advice** 48 **MET** 63 **MET + CBT** 55	*N* = 180 **Control**—*n* = 48 **Brief Advice**—*n* = 37 **MET**—*n* = 55 **MET + CBT**—*n* = 40	ASI BSI NODS DSM–IV SOGS Treatment Service Review	6 weeks 9 months	• Relative to assessment only, brief advice was the only condition that significantly decreased gambling between baseline and Week 6, and it was associated with clinically significant reductions in gambling at Month 9. Between Week 6 and Month 9, MET plus cognitive behavioral therapy evidenced significantly reduced gambling on 1 index compared with the control condition.
Prochaska & DiClemente (1982) [87]	To develop a more integrative model of change: a transtheoretical therapy.	*-*	-	*-*	Transtheoretical therapy	-	• In studying how individuals change on their own compared with change in formalized treatments, four stages of change have been identified. • Individuals changing within and without therapy appear to apply three verbal processes of change in the contemplation and determination stages and then apply two behavioral processes in the action and maintenance stages
Stevens et al. (2013) [96]	To examine the relationship between two validated tasks of decision making and treatment dropout in a relatively large (*n* = 150) and unselected sample of primarily CDI enrolled in long-term residential TCs.	**Treatment completers** *M* = 37.73 *SD* = 8.34 **Dropouts** *M* = 34.87 *SD* = 8.09	**Treatment completers** 94 **Dropouts** 92	*N* = 150 Treatment completers—*n* = 66 Dropouts—*n* = 84	CGT IGT	6 months up until 2 years	• Compared to treatment completers, cocaine-dependent individuals who dropped out of Therapeutic Community prematurely did not establish a consistent and advantageous response pattern as the Iowa Gambling Task progressed and exhibited a poorer ability to choose the most likely outcome on the Cambridge Gamble Task. There were no group differences in betting behavior.
Van Rooji et al. (2012) [93]	Evaluates this pilot treatment program by providing a qualitative analysis of the experiences of the therapists with the treatment of 12 self-proclaimed internet addicts.	*M* = 34.08 *SD* = 12.82	92	*N* = 12	BSCQ CBT MI Compulsive Internet Use Scale	10 weeks	• Therapists report that the program fits the problem of internet addiction quite well. • Therapists further indicated that the treatment achieved some measure of progress for all of the 12 treated patients, while patients reported satisfaction with the treatment and actual behavioral improvements.

Note. ASI = Addiction Severity Index; ASI-G = Addiction Severity Index-Gambling; AUDIT-C = Alcohol Use Disorders Identification Test; BSCQ = Brief Situational Confidence questionnaire; BSI = Brief Symptom Inventory; CBT = Cognitive Behavioral Therapy; CDI = Cocaine-Dependent Individuals; CGT = Cambridge Gamble Task; DASS-21 = Depression Anxiety and Stress Scales; DAST = Drug Abuse Screening Test; DSM-II1 = Diagnostic and Statistical Manual of Mental Disorders; DUDIT = Drug Use Disorders Identification Test; EGM = Electronic Gambling Machines; GHS = Gambling Help Service; GASS = Gambling Abstinence Self-Efficacy Scale; GRCS = Gambling Related Cognitions Scale; GRSEQ = Gambling Refusal Self-Efficacy Questionnaire; GSAS = Gambling Symptom Assessment Scale; GUS = Gambling Urge Scale; I-CBT = Internet-Based Cognitive Behavioural Therapy Program; IGT = Iowa Gambling Task; I-MFS = Monitoring, Feedback, and Support; MI = Motivational Interview; MI + W = Motivational Interview plus Cognitive-Behavioural Self-help Workbook; MI + W + B = Single Motivational Interview plus Workbook plus Four Follow-up Telephone Interviews; NODS = NORC DSM-IV Screen for Gambling Problems; PH1-PRIME-MD = Patient Health Questionnaire; PGSI-CPGI = Problem Gambling Severity Index of the Canadian Problem Gambling Index; PRIME-MD = Primary Care Evaluation of Mental Disorders; PGSI = Problem Gambling Severity Index; PNF = Personalized Normative Feedback; QOLI = Quality of Life Inventory; SOGS = South Oaks Gambling Screen; SWLQ = Satisfaction with Life Questionnaire; TAU = Treatment as Usual; TC = Therapeutic Community.

## 4. Discussion

This review intended to address the following research inquiries: What interventions and methodologies of study have been prominent in gambling research? What implications and recommendations for policy and further research can be derived from these findings to inform people who are in practice? To gather more knowledge on this issue, a systematic literature review was conducted, according to PRISMA guidelines. A total of 13 studies were reviewed.

Research on individuals undergoing self-directed change and those receiving therapy has revealed that clients, like therapists, play an active role in the change process. Therapists who acknowledge that their clients can serve as both catalysts for change and potential obstacles are likely to enhance their effectiveness. One common source of resistance arises when the client and therapist find themselves in different stages of the change process. Therapists who are more directive and action-oriented may encounter resistance when working with clients in the contemplative stage, as these clients may perceive the therapy as progressing too rapidly. Conversely, therapists emphasizing raising awareness may perceive clients ready for action as resistant to the therapeutic process. Clients may be cautioned against impulsive actions, but from their perspective, the therapist may be seen as moving too slowly. This highlights the importance of therapists recognizing and aligning with their clients’ readiness for change to optimize therapeutic outcomes [87].

Various factors, including the escalating prevalence of gambling over recent decades, a limited societal comprehension of gambling disorders, and the perception of gambling as a moral failing rather than a medical challenge, can collectively shape the social acceptability of gambling behavior [100,101,102]. Notably, a significant proportion of individuals in correctional facilities grapple with gambling-related problems (e.g., [103]). It is imperative to incorporate considerations of gambling issues into both treatment and discharge planning for these individuals. Evaluating diverse treatment groups, such as those addressing drug and sex offenses, could provide insights into the presence of gambling-related concerns and relationship patterns. Research focusing on aspects related to access to treatment for individuals with gambling disorders is crucial for understanding their specific needs and enhancing available support systems. Addressing the distinct needs of individuals with gambling disorders holds the potential to increase the number of people seeking and sustaining treatment.

There is a pressing need for public awareness regarding the behaviors associated with gambling addiction and the various treatment options available. Initiatives should be undertaken to enhance the response to the treatment requirements of individuals with gambling disorders and elevate the overall quality of care provided. Key efforts should involve enhancing the professional readiness of therapists and other professionals to effectively assist individuals with gambling disorders. This entails incorporating gambling disorder considerations into diagnostic practices and tailoring treatment plans to address the specific challenges associated with gambling disorders. Furthermore, there is a call to develop therapies that are customized to meet the unique needs of individuals with gambling disorders. Ideally, treatment groups should be designed to include individuals with gambling disorders, facilitating a more comprehensive and targeted approach to addressing their specific concerns.

It is crucial to develop gambling prevention programs tailored for students experiencing academic challenges and dealing with disrupted family relationships. Additionally, there is a need to establish a monitoring system, involving multiple organizations, to track and address unhealthy behaviors related to gambling among adolescents in both school and community settings. Evaluating the extent to which practitioners employ real motivational interviewing techniques during sessions with clients can provide valuable insights for practitioners aiming to facilitate changes in problematic gambling behaviors. Similarly, practitioners are aware that even a single attempt to persuade a client regarding their problematic gambling behavior increases the likelihood of the client resisting change, potentially intensifying issues related to problem gambling and psychological distress. This knowledge holds significant importance for practitioners. In routine care, the tracking of simple motivational interviewing counts can serve as a benchmark for training and supervising practitioners in honing their motivational interviewing skills.

It is intriguing to observe that, aside from the initial study by Prochaska and Di Clemente et al. [87], it was not until Hodgins’ study in 2001 that significant new findings regarding intervention programs on gambling emerged. Remarkably, from that point onward until the present day, there appears to be a notable scarcity of studies aligning with our objectives. This gap spanning two decades may be attributed to the substantial societal transformations that transpired during this period, such as the widespread diffusion of information and communication technology (ICT) [104] and the implementation of diverse policies across various countries [105].

The political sway and potency of the gambling lobby exhibit considerable diversity across nations and regions [106]. Across many locales, the gambling industry exerts substantial influence owing to its economic prowess, marked by revenue generation, job creation, and tourism stimulation. A pivotal element of the gambling lobby’s strength resides in its capacity to financially back political candidates and parties, alongside extensive lobbying endeavors aimed at molding legislation and regulations favorably [107].

Several factors underpin the gambling lobby’s robustness. These include its ample financial resources, which facilitate the funding of political campaigns, the hiring of lobbyists, and the support of advocacy initiatives [107]. Moreover, its role in job creation and economic prosperity often garners favor from policymakers, swaying them to align with its interests. Furthermore, in certain regions, industry consolidation, with a few major corporations dominating the landscape, enables concentrated lobbying efforts for maximal impact [106]. Public support, or successful shaping of public opinion through strategic marketing and public relations campaigns, can also tilt politicians towards alignment with industry interests [108]. Additionally, instances of industry influence over regulatory bodies tasked with oversight further fortify its political clout [105]. Nevertheless, despite its formidable presence, the gambling lobby encounters challenges. Critics raise concerns about the societal and economic toll of gambling, citing issues such as addiction, crime, and adverse effects on vulnerable populations. Moreover, some politicians and policymakers tread cautiously, wary of being perceived as overly cozy with the industry due to ethical considerations or potential public backlash [105].

Although a thorough and systematic search was attempted, using rigorous criteria, there is a possibility that some important studies, due to their inaccessibility, may have not been included. Moreover, it is possible that studies without significant findings were not included in this review, due to the difficulty in publishing these types of results. For these reasons, publication bias is difficult to overcome. Another limitation is the fact that the systematic review was carried out in less than six months and was therefore not registered in PROSPERO, an international database for the registration of systematic reviews in the field of health. The quality of the articles was also not assessed. Considering the defined inclusion and exclusion criteria, only 13 studies were selected. No serious methodological flaws were detected, so it was considered that all of them should be considered. If the article quality assessment grid was applied, some studies might have to be excluded in the process. No restrictions were made regarding temporal or geographic criteria.

Future research endeavors should investigate the impact of reported adherence/non-adherence frequency on the outcomes of motivational interviewing interventions. This focus, alongside the more conventional examination of intervention duration, would provide a more comprehensive understanding of the motivational interviewing dose. Effective strategies, such as online cognitive behavioral therapy and motivational, supportive, and feedback therapy, have demonstrated success in reducing gambling severity (e.g., [97]). To enhance the efficacy of online interventions, further studies should explore methods to boost participant engagement and diminish dropout rates. Additionally, there is a need for research to uncover new ways in which online cognitive behavioral therapy can be utilized to broaden the array of treatment options available for individuals grappling with gambling problems. Given the rising incidence and prevalence of behavioral addictions, particularly in adolescents, it is imperative to delve into a better understanding of this issue. Developing and adapting prevention and treatment plans that cater to gender and age-specific needs is crucial. Furthermore, understanding gender differences in the treatment of behavioral addictions is an important area for exploration.

In conclusion, the research emphasizes the active role that clients play in the process of self-directed change and therapy. Therapists recognizing clients as both catalysts for change and potential obstacles can enhance their effectiveness. A common source of resistance arises when clients and therapists are in different stages of the change process, underlining the importance of therapists aligning with clients’ readiness for change. The prevalence of gambling has increased over recent decades, and societal understanding of gambling disorders remains limited. Gambling is often perceived as a moral failing rather than a medical challenge. Notably, a significant number of individuals in correctional facilities struggle with gambling-related problems. Addressing gambling issues in treatment and discharge planning for these individuals, along with research on diverse treatment groups, is crucial for understanding and meeting their specific needs.

Public awareness regarding gambling addiction behaviors and available treatment options is urgently needed. Efforts should focus on enhancing the response to treatment requirements for individuals with gambling disorders, including improving professional readiness among therapists. Tailoring treatment plans to address the specific challenges associated with gambling disorders and developing customized therapies are essential. Treatment groups should ideally include individuals with gambling disorders for a more comprehensive approach.

## 5. Conclusions

Establishing a monitoring system involving multiple organizations can effectively address unhealthy behaviors related to gambling among adolescents in schools and communities. Practitioners using real motivational interviewing techniques during sessions can provide valuable insights for facilitating changes in problematic gambling behaviors. Recognizing the impact of even a single attempt to persuade a client highlights the need for careful consideration in routine care, using motivational interviewing counts as benchmarks for training and supervision.

## Figures and Tables

**Figure 1 ijerph-21-00346-f001:**
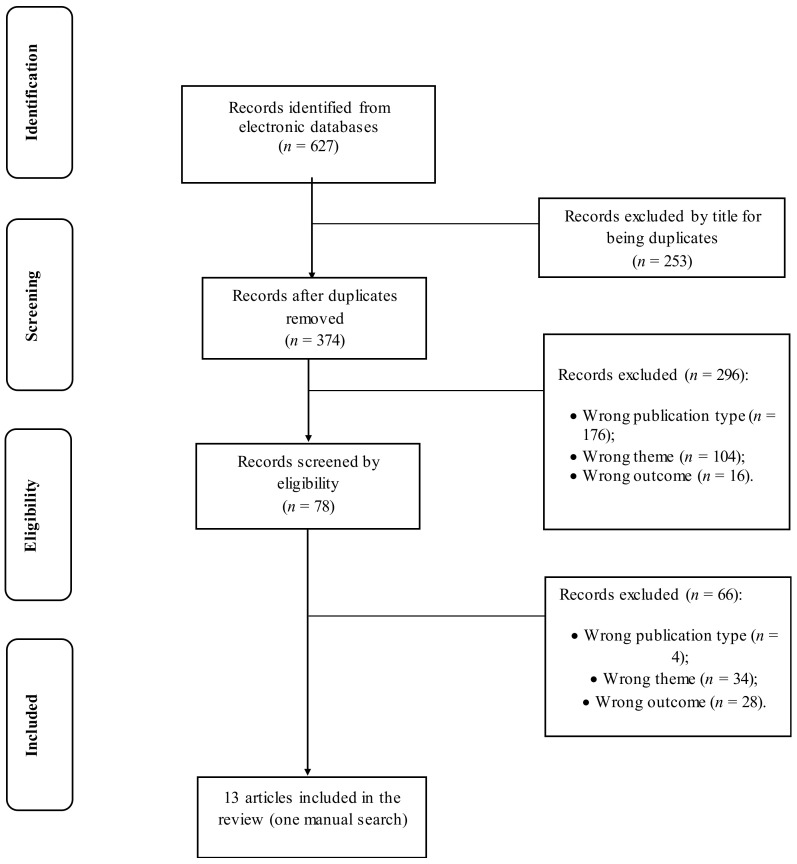
Flowchart of Literature Review Process.

## Data Availability

The data presented in this study are available upon request from the corresponding author.

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
