# Peer review of "A Systematic Review on Intervention Treatment in Pathological Gambling"

_ijerph, 2024, doi:10.3390/ijerph21030346_

Round 1

Reviewer 1 Report

Comments and Suggestions for Authors

Review ijerph-2847442

Thank you for giving me the opportunity to read this interesting manuscript. It provides an overview of worldwide research on pathological gambling based on a systematic review. Overall, the paper is described in an understandable manner and offers insights, but I have major concerns and comments that I hope are helpful for the author.

Title

After reading the paper, I don't think the title is quite appropriate. Perhaps the author can align it more with the goal of the study and the research questions.

Abstract

There are surely more than 13 studies on gambling available, so the abstract should provide more specific details regarding the inclusion criteria. The aim of the study remains unclear in this section.

Introduction

·         I'm missing an overview in the sense of some introductory sentences outlining the content related to gambling that will be discussed here. The various aspects of the studies are currently more straightforwardly listed and could certainly be better organized by the authors.

·         It is striking that very old studies are cited for prevalence figures. After all, 2012 is already more than 10 years ago, and a study from 1999 was used for gender comparison. I would not rely on the validity of these studies' findings and recommend the authors to support these numbers with more recent studies or address the age of the cited studies.

·         Is there a reason why the official criteria for characterization are not utilized at the beginning of the introduction?

·         There are indeed studies on female gamblers.

·         Were existing registries checked for planned/ongoing studies? Was the review itself pre-registered? If not, this constitutes a limitation.

·         Minor comment: In line 72, the citation should be better integrated into the sentence [(2014)'s].

·         Minor comment: The direct quote in line 77 should be immediately followed by a reference citation at the end of the quote.

Materials and Methods

·         The search strings might be more comprehensible if presented in tables.

·         Why was the age of the population set at 12? This number appears somewhat arbitrary.

·         What is the difference between exclusion criteria number 2 and 4?

·         For the title and abstract screening, it was nicely described how many reviewers (and who) were involved - the same information is missing for the full-text screening and data extraction. Who was involved in these stages? Was there also a two-reviewer strategy here? Were the reference lists of the included articles systematically screened? What about the articles that made it to the full-text screening? Were the reference lists also checked here?

·         Was the process conducted using Excel, Covidence, or without any tools?

·         Was there no systematic analysis of the quality of the included studies? If not, I strongly recommend doing so and integrating it into the analysis and summary of the articles.

·         The flow chart should better adhere to the PRISMA criteria (e.g., include details about the hits from individual databases).

Results

·         It would be helpful to initially summarize the results in a clear manner: e.g., “X studies with a total of N=X participants from X countries, investigating X interventions in X settings, were identified”.

·         The results table should be inserted at the beginning of the results section. This way, there is a lack of an overview of which topics each study addressed.

·         I would suggest avoiding numerous abbreviations in Table 1. Additionally, the sources of the used measures are missing.

·         There is a lack of information on how the diagnoses were made in the respective studies, as well as details on how the outcomes were measured and information regarding the settings.

·         Since the introduction referred to possible gender differences, I am wondering about the gender distribution in the included studies. How was the gender balance in the samples, and were there any differences in the effectiveness of interventions?

Discussion

·         A clear summary of the results is missing. What is new? What was already known? There is a lack of connection to the current literature. What is the study's conclusion for research? For practice? For individuals affected?

·         Especially in the context of gambling, I find it relevant to discuss the political dimension and the strength of the gambling lobby.

·         Not a word is mentioned about the quality of the studies. Especially in intervention studies, there are guidelines for assessing quality. This is a significant deficiency.

·         Additionally, there is no section discussing the limitations of the present study, even though there are indeed some limitations.

Author Response

Here it goes the answers.

Reviewer 2 Report

Comments and Suggestions for Authors

Undoubtedly, the present work would be a high-quality systematic review. As we all know, as one of the most considerable issues in the area of behavioral addictions, pathological gambling has been attracting lots of attention all the time. And the present study make a comprehensive and persuasive review about this topic, and also some valuable findings can be attained from this work, and especially that the specific and effective intervention treatments could help us to resolve this troublesome problem afterwards. Given those mentioned above, I strongly recommend this study should be accepted and published as fast as possible, but before conducting that, some detailed issues must be addressed in advance, and the specific advises are listed as follows. Thank you, and good luck to you!

1. The title can be revised as A systematic review on intervention treatment in pathological gambling.

2. The abstract must contain more important contents about the results from systematic review, instead of some irrelevant information.

3. The keywords can be changed as behavioral addiction; pathological gambling; intervention treatment; a systematic review.

4. About table 1, this should be placed at the front of the results.

5. Some contents in conclusion can be put in the section of discussion, and the conclusion should be also reduced and refined. The key contents of the present study need to be displayed here.

6. In addition, the references should be listed by the corresponding standards and norms of the present journal, so this section has been still needed to be carefully and seriously revised by the authors.

Comments on the Quality of English Language

I believe the moderate editing of English language will be required with this work!

Author Response

Here it goes the answers.

Reviewer 3 Report

Comments and Suggestions for Authors

Dear Authors,
thank you for the opportunity to read your interesting paper.

I have just few consideration below:

-  not sure that this is the first review on pathological gambling, please confirm that or change the sentence with one more appropriate such as  "This paper presents a comprehensive overview of worldwide research on pathological gambling."

- Did you use some software or program in your review? if yes, please report in methodology 

-2.3 Identification and screening
....spanning the period from 1931 to 2023.
please report the correct time (e.g from January 1931 to December 2023)

- In sum, except the first study from Prochaska & Di Clemente et al.,
in 1982, we have to wait 2001with Hodgins study to have new findings
regarding intervention programs on gambling, and starting from
this date until today. Would be interesting, in discussion section,
to underlines and to attempt a reason that why in this two decads
there aren't study that reflect your research aims. Probably one
reason could be attempt a lot of changes are occured in societal
habits during this two decades,e.g. ICT diffusion, different
policies in different countries etc.

Author Response

Here it goes the answers.

Round 2

Reviewer 1 Report

Comments and Suggestions for Authors

Thank you for giving me the opportunity to read the revised manuscript. Overall, most of my points were addressed, but there are still some minor comments that I hope are helpful for the author.

Abstract: Line 21: The authors should be more precise: “This paper presents a comprehensive overview of worldwide research on pathological gambling” is not quite correct, as it only refers to interventions.

Introduction: It is still incomprehensible to me why no generally and uniformly valid diagnostic criteria are used according to the current classification systems. Even though the studies cited here are still older, the authors have made an effort to integrate more recent studies than in the previous one, which is very positive.

Materials and Methods: The methods are now described more clearly and comprehensibly, even if there is still room for improvement (in the sense of "who was involved in what").

Author Response

Dear Reviewer 1

You will find our responses and the changes in the text in blue.

Thank you for giving me the opportunity to read the revised manuscript. Overall, most of my points were addressed, but there are still some minor comments that I hope are helpful for the author.

The authors thank you for your careful review of the article considering its improvement.

Abstract: Line 21: The authors should be more precise: “This paper presents a comprehensive overview of worldwide research on pathological gambling” is not quite correct, as it only refers to interventions.

You are right and we have replaced the setence with this one: “This paper provides a comprehensive overview of global research on interventions for pathological gambling”.

Introduction: It is still incomprehensible to me why no generally and uniformly valid diagnostic criteria are used according to the current classification systems. Even though the studies cited here are still older, the authors have made an effort to integrate more recent studies than in the previous one, which is very positive.

The assessment of psychopathology or personality structures, in terms of the intensity with which certain personality and behavioral characteristics are present in an individual, has led to a discussion about the nature of this phenomenon, whether categorical (typological) or dimensional. In the first case, the differences between the individual and other individuals would be qualitative. In the second, they would be quantitative. This question appeared quite early in the empirical tradition (e.g., Hare, 1973). For the typological view, psychopathology would be a taxon, that is, a non-arbitrary class or entity (such as sex or species). Meanwhile, for the dimensional view the characterization of personality structures is defined in terms of a continuum along which all individuals can be arranged. Taxometric analysis research has shown results in favor of both the typological perspective (e.g., Harris et al., 1994; Skilling et al., 2002) and the dimensional perspective (e.g., Guay et al., 2007; Walter et al., 2007; Walters et al., 2007). However, at present, the empirical evidence is more favorable to the dimensional view.

The dimensional conception implies that there are no individuals in the categorical and exclusive sense of the term. Thus, all people can present a greater or lesser degree of personality traits theoretically related to the construct, and in the general population there would be a continuous distribution of these characteristics. The dimensional conception, therefore, admits a certain ambiguity in the characterization of the condition, as it depends on the intensity and scope of the behavioral and personality characteristics that an individual presents. The literature, therefore, has recommended caution in the use of cutoff points to classify individuals as having and not having psychopathology through psychometric instruments (e.g., Walters et al., 2007; Walters et al., 2007). The cutoff points are arbitrary and lack precise empirical justification. Its use, therefore, is potentially dangerous in situations where assessment results have a direct impact on individuals’ lives, such as in forensic and institutional contexts (e.g., Walters et al., 2007; Walters et al., 2007).

Materials and Methods: The methods are now described more clearly and comprehensibly, even if there is still room for improvement (in the sense of "who was involved in what").

The PRISMA guidelines only suggest that we identify the authors who analyze the abstracts. The other authors, along with their respective roles, are identified at the end of the main document.
